# MLN4924 Treatment Diminishes Excessive Lipid Storage in High-Fat Diet-Induced Non-Alcoholic Fatty Liver Disease (NAFLD) by Stimulating Hepatic Mitochondrial Fatty Acid Oxidation and Lipid Metabolites

**DOI:** 10.3390/pharmaceutics14112460

**Published:** 2022-11-15

**Authors:** Mengxiao Ge, Linlin Huang, Yinjun Ma, Shuangyi Sun, Lijun Wu, Wei Xu, Dongqin Yang

**Affiliations:** 1Department of Digestive Diseases of Huashan Hospital, Fudan University, Shanghai 200040, China; 2Key Laboratory of Metabolism and Molecular Medicine of the Ministry of Education, Department of Biochemistry and Molecular Biology, School of Basic Medical Sciences, Fudan University, Shanghai 200032, China; 3Department of Library, Fudan University, 220 Handan Road, Shanghai 200433, China; 4Department of Immunology, School of Basic Medical Sciences, Fudan University, Shanghai 200032, China

**Keywords:** non-alcoholic fatty liver disease (NAFLD), MLN4924, neddylation, mitochondrial fat acid oxidation, lipid metabolism

## Abstract

MLN4924 is a selective neddylation inhibitor that has shown great potential in treating several cancer and metabolic diseases, including obesity. However, it remains largely unknown whether MLN4924 has similar effect on non-alcoholic liver disease (NAFLD), which is closely associated with metabolic disorders. Here, we investigated the role of MLN4924 in NAFLD treatment and the underlying mechanism of the action using primary hepatocytes stimulated with free fatty acid, as well as high-fat diet (HFD)-induced NAFLD mouse models. We found that MLN4924 can inhibit the accumulation of lipid and reduce the expression of peroxisome proliferator-activated receptor γ (PPARγ), a key player in adipocyte differentiation and function in both in vivo and in vitro models. Moreover, we verified its important role in decreasing the synthesis and accumulation of fat in the liver, thus mitigating the development of NAFLD in the mouse model. The body weight and fat mass in MLN4924-treated animals were significantly reduced compared to the control group, while the metabolic activity, including O_2_ consumption, CO_2_ and heat production, also increased in these animals. Importantly, we demonstrated for the first time that MLN4924 can markedly boost mitochondrial fat acid oxidation (FAO) to alter liver lipid metabolism. Finally, we compared the metabolites between MLN4924-treated and untreated Huh7 cells after fatty acid induction using lipidomics methods and techniques. We found induction of several metabolites in the treated cells, including Beta-guanidinopropionic acid (b-GPA) and Fluphenazine, which was in accordance with the increase of FAO and metabolism. Together, our study provided a link between neddylation modification and energy metabolism, as well as evidence for targeting neddylation as an emerging therapeutic approach to tackle NAFLD.

## 1. Introduction

Liver, an essential organ tightly related to lipid metabolism, is responsible for both the regulation of lipid homeostasis and utilization of energy [1]. As the most rapidly increasing cause of liver-related mortality, NAFLD has become prevalent worldwide, estimated to affect up to 25% of the global adult population. NAFLD histologically encompasses non-alcoholic fatty liver (at least 5% steatosis with or without mild inflammation) and non-alcoholic steatohepatitis (5% hepatic steatosis and inflammation with hepatocyte injury) [2,3]. Despite large amounts of ongoing research on the pathogenesis and drug development of NAFLD, no specific drug has been developed yet to treat NAFLD.

NAFLD is strongly associated with obesity and metabolic syndrome, conditions that cause lipid accumulation in hepatocytes, which, in part, reflect malfunction of the adipose tissue, the long-term lipid storage organ in the body [4]. Brown adipose tissue (BAT) and white adipose tissue (WAT) are the two main types of adipose tissues in the body, and have antagonistic functions. BAT is responsible for adaptive thermogenesis [5] while WAT is the primary site for energy storage in the form of triglycerides, the dysfunction of which is associated with obesity [6]. Adipose tissue can impact hepatic cell function by the production of cytokines and hormones, as well as by regulating lipid flux to the liver. It was shown that WAT hyperplasia may contribute to the development of NAFLD through systemic insulin sensitivity regulation [7]. BAT was also found to be involved in hepatic metabolism by various mechanisms, including decreasing the levels of circulating fatty acids, secreting different cytokines and crosstalk with other related organs [8]. As liver serves as an endocrine organ that regulates metabolic processes in both liver and non-hepatic tissues by secreting hepatokines, dysfunction of the liver can also induce systemic metabolic disorders [9].

The maturation of adipocytes is a complex process that involves multiple pathways on both transcriptional and post-transcriptional levels [10]. Neddylation is a post-transcriptional modification in which small molecule neuronal precursor cell-expressed developmentally downregulated protein 8-activating enzyme (NEDD8) is covalently added to a substrate protein, thus regulating the structure and function of the target proteins. NEDD8, as an important modifier of ubiquitination, can be induced during the differentiation of preadipocytes and plays an important role in the maturation of adipocytes and accumulation of lipid. Recent studies have shown that inhibition of neddylation can alleviate steatosis [11], reduce liver fibrosis [12] and inhibit cancers, including esophageal cancer [13], metastatic melanoma [14], acute myeloid leukemia [15], lymphoma and multiple myeloma [16]. MLN4924 is a highly selective neddylation inhibitor. It blocks the neddylation modification by inhibiting NEDD8-activating enzyme (NAE), which leads to the inactivation of cullin-RING ligases (CRLs) [17,18]. MLN4924 has been reported to regulate apoptosis, autophagy, cell cycle arrest, DNA repair and replication [19]. It was also shown to suppress tumorigenesis in multiple human cancers [20] and, therefore, has been included into some clinical trials as an anticancer agent alone or in combination with various chemotherapeutic drugs [21].

In addition to its anti-tumor effect, MLN4924 was recently shown to prevent adipogenesis and lipid droplet formation in 3T3-L1 cells [22]. It can also suppress obesity and glucose intolerance induced by HFD in mice [23]. We, therefore, speculate the inhibitory effect of the neddylation inhibitor MLN4924 on the development or progression of NAFLD and analyze its value in treating metabolic diseases. To test our hypothesis, we quantified the contents of lipid droplets in hepatocytes and determined conditions of all types of the adipose tissues and liver in the HFD-mice with or without MLN4924. We observed a marked increase in heat production and elevated hepatic FAO activity in HFD-mice treated with MLN4924 compared to the control mice. To investigate the mechanism by which neddylation inhibitor MLN4924 regulates lipid metabolism and alleviates HFD-diet induced NAFLD, we used untargeted metabolomics to identify metabolites and relevant pathways in Huh7 cells after MLN4924 treatment. We identified b-GPA and Fluphenazine, which was in accordance with the increase of FAO and lipid metabolism.

## 2. Materials and Methods

### 2.1. Animals

The male C57BL/6J mice (6–8 weeks of age) were obtained from Shanghai Jihui Laboratory Animal Care Company and housed in a pathogen-free facility with a 12 h light/12 h dark cycle at 22–25 °C. The male C57BL/6J mice were randomized into 4 groups (5–6 animals per group). A vehicle or MLN4924 (30 mg/kg) was injected into mice intraperitoneally twice a week for 12 weeks. For diet challenge, two groups (vehicle or MLN4924 given) were fed a diet containing 60% fat, 20% carbohydrates and 20% protein (D12492, Research Diets). Body weight was measured once a week. After 12 weeks, blood, liver, iWAT (inguinal white adipose tissue), gWAT (gonadal white adipose tissue) and BAT (brown adipose tissue) samples were collected and weighed. The right posterior lobe of the liver and part of the adipose tissue samples were fixed in 4% formaldehyde and the remaining samples were kept in liquid nitrogen. All animal experiments were previously approved by the Animal Ethics Committee of Huashan Hospital.

### 2.2. Cell Culture and Lipid Droplet Induction

The human liver cancer cell line Huh7 cells were obtained from Cell Bank of Chinese Academy of Medical Science (Shanghai, China). Cells were cultured in Dulbecco’s modified Eagle’s medium (11966025, Thermo Fisher Scientific, Waltham, MA, USA) supplemented with 10% fetal bovine serum (FBS) (12484010, Thermo Fisher Scientific, Waltham, MA, USA) and 1% penicillin-streptomycin (PS) at 37 °C with 5% CO_2_. 6.66 mM OA (Oleic acid) (O7501, Sigma-Aldrich, USA) and 3.33 mM PA (Palmitic acid) (P9767, Sigma-Aldrich, St. Louis, MO, USA) were prepared, respectively, with 10% fat acid-free BSA (A1595, Sigma-Aldrich, St. Louis, MO, USA) solution. According to a 1:10 proportion, OA and PA were mixed into DMEM with 10% FBS to prepare the high fat medium with 1 mM FFA. A vehicle or MLN4924 was added into the high fat medium 2 h in advance.

### 2.3. Western Blotting

The cells or tissues were lysed by RIPA buffer (P0013C, Beyotime, Shanghai, China) with phenylmethanesulfonyl fluoride (PMSF) (ST506, Beyotime, Shanghai, China) and protease cocktail (B15002, Bimake, Houston, TX, USA). The proteins in the lysates were electrophoresed and separated on SDS-polyacrylamide gels, transferred to polyvinylidene fluoride membranes and detected by antibodies that recognize PPARγ (#2435, CST) and *GAPDH* (*#5174*, *CST*), The bands were visualized using Chemiluminescent HRP Substrate (WBKLS0500, Millipore, Burlington, MA, USA) and imaged by the ImageQuant LAS (GeHealthcare, Chicago, IL, USA).

### 2.4. Mitochondria, Bodipy and DAPI Staining

The cell samples were washed by PBS 3 times. The 10 mM bodipy dye (D2184, Thermofisher Scientific, Waltham, MA, USA) was diluted to a 2-μM working solution according to the proportion of 1:5000. The 10 mM Mitochondria dye was diluted into 2 μM working solution according to the proportion of 1:2000. The samples were stained by working solution and incubated in a 37 °C incubator for 15 min and washed by PBS 3 times, then fixed with formaldehyde (P0099, Beyotime, Shanghai, China) for 30 min at 4 °C. DAPI (C1002, Beyotime, Shanghai, China) solution was diluted according to the proportion of 1:5000. The samples were photographed using a fluorescence microscope (Leica biosystems, Wetzlar, Germany).

### 2.5. Isolation of Mouse Primary Hepatocyte

F12 medium was prepared with 1% PS, insulin (10 mol/L), dexamethasone (10 mol/L) and 10% FBS. Hanks (H1025, Solarbio, Beijing, China) was prepared with 0.25 g/L collagenase IV (C5138, Sigma-Aldrich, St. Louis, MO, USA). The 40% percoll solution was prepared with 10× PBS, percoll and F12 medium (17-0891-01, GE Healthcare, Chicago, IL, USA). The procedure was performed under anesthesia. The tube and needle were fixed on the peristaltic pump and filled with 37 °C D-Hanks (H1045, Solarbio, Beijing, China). The 26G needle was inserted into the hepatic portal vein. The pump was turned on and the warm perfusion buffer was allowed to reach the needle. Then, the inferior vena cava was cut. The liver was perfused continuously to wash out blood and circulating cells until the liver became large and pale. Then, the collagenase IV was perfused to the liver in order to digest collagen in the extracellular matrix, thereby facilitating cell dispersion. The liver was dissected out carefully to a dish filled with 10 mL cold F12 medium at 4 °C, with the gallbladder removed. The F12 medium was filtered through a 100 μM cell strainer into a 50 mL collection tube. The cell suspension was centrifuged at 50× *g* for 3 min at 4 °C and the supernatant was removed. The precipitate was resuspended with percoll solution and the suspension was centrifuged at 150× *g* for 8 min at 4 °C. The supernatant was removed and the cells were suspended with F12 medium. The primary hepatocytes were maintained at 37 °C with 5% CO_2_ in a humidified environment.

### 2.6. Liver Triglyceride Extraction

Cells were collected and the intracellular triglycerides were estimated using a triglyceride assay kit (E1013-50, Applygen Technologies Inc., Beijing, China) according to the manufacturer’s recommended protocol. Lysate buffer was added and allowed to stand for 10 min. The supernatant was heated at 70 °C for 10 min. The supernatant was colleted after centrifugation at 12,000× *g* at 20 °C for 10 min. R1 and R2 working solutions were mixed according to the proportion of 4:1. The standard triglycerides samples were diluted and a blank tube was set as control. Measurements of 10 μL of supernatant and 190 μL of working solution were mixed to react for 15 min at 37 °C. The absorbance was measured at 550 nm.

### 2.7. Small Animal NMR Analyzer and Metabolic Cages Measurements

We assessed the fat and lean masses of live mice using an NMR analyzer (Minispec Live Mice Analyzer (LF50), Bruker, Billerica, MA, USA) according to the manufacturer’s instructions. The mice were then put into the Comprehensive Lab Animal Monitoring System (CLAMS, Columbus Instruments, Columbus, OH, USA) to record O_2_ consumption, CO_2_ production, energy expenditure and food intake. Animals were acclimatized in the recording chambers for 24 h, and measurements were taken subsequently for 72 h during the light cycle and dark cycle with free access to food and water. VO_2_, VCO_2_ and energy expenditure were normalized to the lean mass of the mice.

### 2.8. Fatty Acid Oxidase Testing

Liver tissues were collected and treated with the Fatty Acid Oxidation Assay Kit (CB11345-Mu, COIBO BIO Inc., Shanghai, China) according to the manufacturer’s recommended protocol. Briefly, liver tissues were isolated from MLN4924-injected mice and control mice after a 12-week HFD. The tissues were weighed and minced to small pieces into a tube. Then, the tissues were incubated in 180 μL PBS containing protease cocktail at 4 °C after homogenization for 10 min. The cell suspension was centrifuged at 5000× *g* for 5 min at 4 °C and 50 μL supernatant was added into the 96-well plate. After adding 150 μL enzyme-conjugate, the mixture was incubated at 37 °C for 15 min. Finally, 50 μL substrate A and 50 μL substrate B were added to each well. Then, the substrates were mixed gently and incubated for 15 min at 37 °C. An amount of 50 μL stop solution was added to each well. The optical density was recorded at 450 nm using a microtiter plate reader.

### 2.9. Statistics

All statistics were are presented as mean ± SD. *p* < 0.05, *p* < 0.01 and *p* < 0.001, as three levels were all considered to be significant for a difference. The two-tailed unpaired Student’s *t* test was used to compare two groups. One-way ANOVA was used for comparison among 3 or 4 groups and two-way ANOVA was used for body weight analysis, all followed by Turey test. Statistical analyses were performed using GraphPad Prism 9.3.1 software.

## 3. Results

### 3.1. Prolonged Exposure to Free Fatty Acid Can Induce Lipid Accumulation in Primary Hepatocytes

To determine the effect of lipid accumulation, primary hepatocytes were treated with free fatty acid (FFA) for various lengths of time and were stained for DNA (DAPI in blue) and lipid droplets (BODIPY in green) post-treatment (Figure 1a). In cells with prolonged treatment, the intensity of both green fluorescence and oil red o (ORO) staining increased significantly, and larger lipid contents were easily observed (Figure 1a,b). Quantification of lipid droplets, either by relative levels (represented by ratio of BODIPY over DAPI positive area) or the percentage of ORO positive area, confirmed increased lipid accumulation in the cells with prolonged treatment (Figure 1c,d). Huh7 cells also showed a similar trend with accumulated lipid droplets after treatment and staining with BODIPY and ORO (Appendix A). Moreover, it was found that the BODIPY staining exhibited strong co-localization with mitochondrial staining (in red) (Appendix A), indicating that mitochondria were involved in lipid accumulation in these cells.

PPARγ was reported to play an important role in lipid synthesis and accumulation as a lipid-forming hormone receptor [24]. The expression level of PPARγ is the highest in adipose tissue, where it functions as an inducer of adipocyte differentiation and modulates an array of target genes that are involved in lipid uptake and storage [25]. Hepatic expression of PPARγ promoted an adipogenic transformation of hepatocytes resulting in an increase in hepatic triglyceride levels [26]. In agreement with previous findings, we observed increased PPARγ expression in both Huh7 cells and primary hepatocyte upon FFA treatment (Figure 1e,f).

### 3.2. Neddylation Inhibitor MLN4924 Can Prevent Lipid Accumulation Induced by FFA in Hepatocytes

MLN4924 is a newly discovered synthetic compound that inhibits the neddylation process and neddylation modification plays a critical role during differentiation of 3T3L1 cells into adipocytes [27]. To determine whether MLN4924 regulates lipid metabolism in isolated hepatocytes cell models of steatosis, we treated primary hepatocyte with MLN4924 at different concentrations 2 h before addition of FFA. After 8-h FFA treatment, lipid droplet accumulation was determined by BODIPY staining in the cells (Figure 2a). The green fluorescence intensity gradually decreased in both size and number of lipid droplets in a MLN4924-dose-dependent manner. Oil red staining was used to measure the size and number of lipid droplets in primary hepatocytes (Figure 2b), and statistical analysis further confirmed that reduced lipid droplet accumulation in liver cells was strongly correlated with increased dose of MLN4924 (Figure 2c,d). Regardless of the concentration of MLN4924, both mitochondria (red fluorescence) and lipid droplets (green fluorescence) staining co-localized in Huh7 cells (Appendix A). These data demonstrated that MLN4924 can inhibit the accumulation of lipid in primary hepatocytes. However, whether MLN4924 inhibits lipid accumulation by reducing lipid synthesis or increasing metabolic cost remains to be further investigated.

PPARγ neddylation, leading to stabilization of the protein, is essential for both adipogenesis and fat accumulation, and PPARγ was reported as one of the neddylated proteins in 3T3-L1 derived adipocyte-like cells [23]. Consistent with previous findings, we also found that the expression levels of PPARγ in primary hepatocytes and Huh7 cells were negatively correlated with the concentration of MLN4924 (Figure 2e,f), suggesting that the anti-steatotic effect of MLN4924 was likely via inhibition of neddylation.

### 3.3. MLN4924 Treatment Protected HFD-Fed Mice from Developmenting of NAFLD and Obesity

To investigate the effects of MLN4924 on lipid metabolism in vivo, C57BL/6J mice were randomly divided into four groups to receive different treatment and referred to as HFD-MLN4924, HFD-Control, NCD-MLN4924 and NCD-Control. During the feeding period, mice from all four groups were weighed weekly. After 3 weeks of feeding, a notable difference in body weight was observed between the HFD-Control and HFD-MLN4924 groups, and animals from HFD-MLN4924 on average weighed 1.95 g (*p* = 0.0495) less than those from the HFD-Control group, with the mean different percentage reaching 7.08% (Figure 3a). The body weight difference reached maximum to 6.67 g (*p* = 0.0005) and the percentage reached to 18.62% after 9 weeks of feeding (Figure 3a). At the 12 weeks feeding endpoint, the difference of body weight between the HFD-Control and HFD-MLN4924 groups was up to 4.86 g (*p* = 0.0052) and the percentage of body weight difference was 14.40% (Figure 3b). In contrast, there was no statistical difference in the weights between the NCD-Control and NCD-MLN4924 groups, indicating that MLN4924 does not influence the body weight and metabolism of animals under a normal diet (Figure 3a). Dysfunctional hepatic lipid metabolism is a cause of NAFLD and was reported to be closely related with insulin resistance and type 2 diabetes [28]. Insulin resistance is involved in the progression of steatosis [29]. To observe the effects of MLN4924 on insulin resistance, we performed glucose and insulin tolerance tests in the HFD-induced mice. We found that mice from the HFD-MLN4924 group exhibited significantly improved glucose tolerance and showed an increased level of insulin sensitivity compared to those from the HFD-Control group (Appendix A), although not to the level of statistical significance. The peak value was reached between 15 to 30 min after glucose injection in both groups, while throughout the whole process, the mean values of the blood glucose in the mice of the HFD-Control group were all higher than that in the mice of the HFD-MLN4924 group (Appendix A).

After 12 weeks of feeding, fat mass, lean mass and fluid mass of the mice from all four groups were measured by the animal NMR analyzer. The average fat mass of mice from the HFD-MLN4924 group was only 2.38 g, which was about 5.28 g less (*p* < 0.001) than those from the HFD-control group (Figure 3c). Whereas no differences in fat mass, lean mass or fluid mass were observed between the NCD groups (Figure 3c). At the end point of feeding, the mice were sacrificed for further analysis. The weights of liver, iWAT, gWAT and BAT from all the animals were first weighed and analyzed (Figure 3d–k). The size of the livers from HFD-fed mice were larger than those from NCD-fed mice, which was consistent with previous reports [30]. The average weights of liver, iWAT and gWAT from mice in the HFD-MLN4924 group were all markedly decreased compared to those from the HFD-control group (15.63%, 47.29%, 62.27%, respectively) (Figure 3d–f). The percentages of iWAT or gWAT weight in total body weight were also lower in mice from the HFD-MLN4924 group, with average iWAT and gWAT weight percentage of body weight being 0.77% vs. 1.22% (*p* = 0.0031), 1.45% vs. 3.21% (*p* = 0.0002) in animals from the HFD-MLN4924 and HFD-control groups, respectively (Figure 3i,j). In contrast, there were no significant differences in the weights of BAT tissue across all four groups (Figure 3g,k). In conclusion, MLN4924 primarily regulates lipid metabolism in liver and white adipose tissue while whether it has effects on brown adipose tissue remains to be explored.

### 3.4. MLN4924 Limits Lipid Accumulation and the Size of Lipid Droplets in Metabolic Tissues

To determine whether MLN4924 has an effect on metabolic tissues, histology examination was performed on liver and adipose tissues. It was found that there was almost no fat accumulation in the liver of mice from the HFD-MLN4924 group, while there were a number of white vacuoles in the liver of mice from the HFD-Control group, indicating that some hepatocytes had undergone ballooning degeneration (Figure 4a). These results indicated that MLN4924 played an important role in preventing or relieving hepatic steatosis. In addition, H & E staining of BAT exhibited smaller adipocytes with a high density of intracellular organelles from mice of the HFD-MLN4924 group compared to that from the HFD-Control group (Figure 4a), suggesting MLN4924 treatment may induce active BAT and a subsequent increase in energy expenditure in animal models [31]. Next, the size of adipocyte was measured using histological sections of iWAT and gWAT tissues of mice from all four groups. An area of 100 random adipocytes from each sample was measured and the statistical distribution map of adipocyte size in animals from the four groups was obtained (Figure 4b,c). The adipocyte size of gWAT and iWAT from the HFD-MLN4924 group was much smaller than that from the HFD-Control group, and was comparable to that from the NCD groups (*p* < 0.001), indicating that MLN4924 could reduce the accumulation of lipid in adipose tissues in mice (Figure 4d,e).

### 3.5. MLN4924 Promotes O_2_ Consumption, CO_2_ Production and Thermogenesis in HFD-Induced NAFLD Mice Model

To determine whether reduced body weight and fat mass in the HFD-MLN4924 group was the consequence of increased metabolism, the metabolic rate of the mice from all four groups were measured. It was found that the daytime O_2_ consumption, CO_2_ production and heat production of mice from the HFD-MLN4924 group were significantly higher than those from the HFD-Control group, while the respiratory quotients were comparable between the two groups (Figure 5a–d). The nocturnal O_2_ consumption, CO_2_ production and heat of mice from the HFD-MLN4924 group showed a trend of increase compared with those from the HFD-Control group, although there were no statistical differences (Figure 5a–d). These results demonstrated that neddylation inhibitor MLN4924 enhanced lipid consumption in HFD-fed mice, resulting in a lean phenotype and reduced accumulation of fat in the liver in these animals. This finding was also consistent with a previous report, which underscored the importance of MLN4924 in ameliorating lipid accumulation in mice [23].

### 3.6. MLN4924 Inhibits Neddylation, Elevates Hepatic Fatty Acid Oxidase Levels, and Reduces Hepatic Triglyceride Levels in HFD-Induced NAFLD Mice Model

To investigate the underlying mechanisms by which MLN4924 modulates lipid metabolism, we first examined whether neddylation was affected in hepatocytes of animals from the HFD-MLN4924 group by immunohistochemistry analysis of NEDD8 in liver tissue. Expression of NEDD8 in hepatocytes can be elevated by HFD, whereas MLN4924 treatment reduced the levels of NEDD8 (Figure 6a). As one of the key players in the maintenance of adipose tissue, PPARγ is subject to various posttranslational modifications, including ubiquitination in adipocytes as well as in hepatocytes [23]. Immunohistochemistry analysis of PPARγ revealed that the expression of PPARγ decreased in the liver of MLN4924-treated mice, which was consistent with the findings of reduced lipid metabolism in these animals (Figure 6a). Notably, we observed an increase in the expressions of carnitine palmitoyltransferase 1A (CPT1A) in the liver tissue of MLN4924-treated mice by the immunohistochemistry analysis. CPT1A, one of the key enzymes in FAO, can revert HFD-induced obesity and NAFLD in mice by enhancing liver FAO [32]. Staining of lipid droplets with ORO further confirmed that the accumulation of lipid droplets in the liver of mice from the HFD-MLN4924 group was much reduced compared to that from the HFD-Control group (Figure 6b). We then measured the levels of intracellular triglycerides to determine the effect of MLN4924 on lipid contents in hepatocytes. The triglycerides levels in hepatocytes from HFD-fed mice were significantly higher than those from HFD-fed mice treated with MLN4924 (Figure 6c). We also examined the levels of fatty acid oxidase in hepatocytes and found that FAO level increased after MLN4924 treatment. Together, these results suggested that inhibition of neddylation by MLN4924 can boost lipid metabolism by increasing FAO levels in the liver (Figure 6d).

### 3.7. Lipidomics Analysis Uncovered Differences in Lipid Metabolites from Liver Cells Treated with MLN4924

Recent studies showed that different lipid metabolites can impact the process of lipid metabolism. To test whether MLN4924 treatment altered the variety or level of the lipid metabolites in hepatocytes, the metabolites of Huh7 cells treated with FFA + DMSO or FFA + MLN4924 were extracted and lipid metabolomics analysis was performed using Liquid Chromotography with Mass Spectrometry techniques (Figure 7a). Partial least squares discriminant analysis (PLS-DA) was applied to the samples data. Either in the positive ion model (left) or the negative ion model (right), the intra-group differences between samples were small, and each data point within the same sample group clustered together, indicating a good reproducibility of the samples (Figure 7b). The contributions of the first component (PC1) and the second component (PC2) to the samples were 30.5% and 31.3% respectively in the positive ion mode, and were 36.6% and 8.7% respectively in the negative ion mode (Figure 7b). Meanwhile, there were the two sample groups clustered separately indicating that MLN4924 treatment could affect cell metabolism. According to the types of metabolites, the accumulation patterns of metabolites among the samples were analyzed by cluster analysis (Figure 7c). The metabolites profile in samples from the same group were comparable. In the negative ion mode, the profiles of metabolites between the two groups differ greatly, which were displayed on both sides of the cluster map, suggesting that MLN4924 could modulate hepatocytes lipid metabolism.

A total of 5370 known metabolites were detected in positive ion mode, and a total of 2758 known metabolites were detected in negative ion mode (Figure 7d). In both modes, there were substantial metabolic changes in MLN4924-treated group compared to control group (Figure 7d). In the positive ion model, 46 differentially produced metabolites were found between the two groups, among which 23 metabolites were up-regulated and the other 23 metabolites were down-regulated in the MLN4924-treated group compared to that in the control group. In the negative ion mode, 31 differential metabolites were detected between the two groups, among which 2 metabolites were up-regulated and 29 metabolites were down-regulated in the treated group. The significant differences in metabolites between the control group and MLN4924-treated group were annotated to KEGG database, and the main metabolic pathways involved were identified by PATHWAY analysis. In the negative ion mode, 31 differential metabolites were enriched between the two groups, and significant changes took place in peptidoglycan synthesis, histidine metabolism, D-alanine metabolism, cAMP resistance and other key pathways (Figure 7e).

Analysis of the top 30 differentially produced metabolites between FFA + DMSO treated group and FFA + MLN4924 treated group (listed in Table 1) uncovered several metabolites that were previously reported to be closely linked to metabolic enhancement which may explain increased FAO in liver upon MLN4924 treatment. For example, b-GPA, a creatine analog, can inhibit creatine uptake by the cell and b-GPA treatment was demonstrated to raise basal metabolic rate and hypoxic exercise tolerance in mice model [33]. Fluphenazine and related drugs were reported to suppress mitochondrial fatty acid oxidation in mouse heart and liver by inhibiting the function of mitochondrial carnitine palmitoyltransferase, cytochrome c oxidase and peroxisomal carnitine octanoyltransferase [34]. Moreover, we found that alteration of some metabolites induced by MLN4924 treatment, such as uridine, may account for enhanced systemic metabolism. Uridine supplementation has been shown to reduce the body weight and the lipid accumulation in the liver, iWAT and gWAT in HFD-fed mice [35]. Taken together, we identified several metabolites that were regulated by MLN4924 in liver and were involved in the regulation of lipid metabolism.

## 4. Discussion

In the current study, we investigated the role of neddylation inhibitor MLN4924 in the development and progression of the HFD-induced NAFLD model. Primary hepatocytes stimulated with FFA and NAFLD mouse models induced by HFD were treated with MLN4924 and both exhibited the same phenotype—reduced lipid accumulation and increased lipid metabolism. This observation indicated that steatosis can be pharmacologically controlled through inhibition of the neddylation. In addition, we found that MLN4924-treated mice under HFD showed increased liver FAO and, importantly, alleviated manifestation of NAFLD, improved O_2_ consumption, CO_2_ production and thermogenesis compared to the HFD-fed control mice. We further explored the underlying mechanism by performing lipid metabolomics analysis on the liver cells. Lipidomics data showed that levels of some metabolites involved in lipid metabolism pathways changed significantly upon MLN4924 treatment. Among these metabolites, we identified a special compound, fluphenazine, which was reported to inhibit FAO by regulating carnitine palmitoyltransferase activity [34]. These changed levels of metabolites could explain, at least in part, the enhanced hepatic FAO and ameliorated steatosis observed in MLN4924-treated mice with HFD-induced NAFLD.

Recent studies have demonstrated the importance of neddylation, a ubiquitin-like posttranslational modification in the process of chronic liver diseases progression, and yet, the underlying mechanisms were complicated and largely unknown [36]. Studies have reported that patients with hepatocellular carcinoma, intrahepatic cholangiocarcinoma, as well as mouse models of liver fibrosis all showed a notable increase in the global neddylated-proteome and neddylation intermediates [12,37]. In 2009, Soucy et al. first demonstrated a small molecule inhibitor of neddylation, MLN4924 (pevonedistat) disrupts cullin-RING ligase-mediated protein turnover [38]. It was reported that MLN4924 can trigger cell cycle arrest, apoptosis and senescence in cancer cells through blocking cullin neddylation, and inactivates CRL, which results in the accumulation of CRL substrates [19,39,40]. Pre-clinical-based research also provided strong evidence that neddylation inhibition is potentially an effective and safe therapeutic target in the context of drug development [36,41]. Previous findings showed that the levels of NEDD8 protein were increased in patients with hepatic steatosis compared to those in healthy controls. Oxidative stress, one of the key pathological features in the progression of NAFLD, could be related to the activation of neddylation [42,43,44]. Interestingly, our results showed that MLN4924 almost reversed the phenotypes of NAFLD induced by HFD, namely histological steatosis (Figure 4a), compared with mice fed with NCD. However, the effects of MLN4924 on non-alcoholic steatohepatitis (NASH) have not been reported and the relationship between neddylation and inflammation in NASH still remained to be discovered. Nevertheless, in our model, we found that after 12 weeks of HFD or NCD feeding, the liver tissues of mice injected with or without MLN4924 showed no pathological changes of fibrosis (Appendix A). Additionally, the levels of macrophage infiltration indicated by F4/80 staining in the liver of animals from the HFD-MLN4924 group were lower than that from the HFD-Control group, indicating that MLN4924 could alleviate the hepatic inflammation caused by HFD (Appendix A).

The transcription factor sterol regulatory element-binding protein 1c (SREBP1c) mainly regulates hepatic lipogenesis [45,46]. Additionally, the PPARγ signaling pathway, which starts and persists during lipid storage, powerfully drives the complex biological process [23]. Neddylation enhances SREBP1c-dependent hepatic lipogenesis and stabilizes PPARγ by competing with ubiquitination [23,46]. Uk-Il Ju et al. showed that the blockade of SREBP1c neddylation could attenuate HFD-induced hepatic steatosis. We also investigate the relationship between neddylation and SREBP1c by examining the level of SREBP1c in the liver of animals from the NCD-fed and HFD-fed groups. We found that, consistent with the reported study, the expression of SREBP1c was reduced in the MLN4924-injected mice compared with that in the Control groups (Appendix A) [46]. Therefore, neddylation research has inspired new ideas for the treatment of non-alcoholic fatty liver disease.

Here, we demonstrated that lipid droplets in hepatocytes exhibited strong co-localization with the mitochondrion, and inhibition of neddylation by MLN4924 can enhance lipid metabolism in the liver via increasing mitochondrial FAO levels. It was reported that inhibiting neddylation modification can alter mitochondrial morphology and reprogram energy metabolism by promoting mitochondrial fusion and the oxidative phosphorylation system (OXPHOS) in cancer cells [47]. Mitochondria play an essential role in cell death, energy production, metabolism and other cellular activities [48,49]. Maintenance of mitochondrial health involves an expanding array of biological processes at both molecular and organelle levels [50]. Mitochondria are also tightly involved in systemic lipid metabolic disorders, thus affecting the occurrence and development of various metabolism-related diseases, including hypertension and cardiac hypertrophy [51]. Many studies have shown the strong connection between mitochondria and lipid metabolism, most of which focused on how the mitochondria function in liver health and metabolic diseases ranging from obesity, NAFLD, to hepatocellular carcinoma [52,53]. Adipocytes can use mitochondrial oxidative machinery and effectors to regulate the rate of substrate oxidation and the heat production [54]. Mitochondria can control lipid generation and adipokine secretion in adipocytes by its key role in substrates and energy production in cell metabolism [55,56,57]. These are consistent with our findings, which implied that mitochondrion can be targeted to reduce liver fat accumulation.

The occurrence and development of NAFLD is closely related to the synthesis and accumulation of lipid in liver tissue. The excess fat in the liver can be reduced by either increasing lipid metabolism or decreasing lipid synthesis. Mitochondria are also closely related to systemic lipid metabolic disorders, thus affecting the development and progression of various metabolism-related diseases, including hypertension and cardiac hypertrophy [42,49,53,56]. FAO, which takes place in mitochondrion, has been considered to be the primary mechanism of fatty acids degradation. Defective FAO can confer more susceptibility to cell dysfunction and has been postulated as one of the mechanisms that contribute to obesity-related metabolic disorders and lipid accumulation [58]. Previous studies showed that hepatic neddylation can facilitate fatty acid β-oxidation through the DEPTOR–mTOR axis (DEP-domain containing mammalian target of rapamycin (mTOR) interacting protein-mTOR) or via targeting and stabilizing electron transfer flavoproteins [58]. Our results showed that neddylation inhibition in HFD-fed mice leads to increased expression of DEPTOR and decreased levels of NEDD8 (Appendix A) [58]. Furthermore, we observed higher O_2_ consumption, CO_2_ production and heat in the HFD-MLN4924 group compared to the HFD-Control group during the daytime, and a higher level of CPT1A (a rate-limiting enzyme of FAO specifically expressing in the liver) in both the HFD-MLN4924 group and NCD-MLN4924 group compared to the Control groups. Fluphenazine was one of the most down-regulated metabolites in MLN4924-treated samples. It has been reported to inhibit mitochondrial FAO by impairing carnitine palmitoyltransferase activity [34,59]. Therefore, fluphenazine seems to be an interesting candidate that mediates the critical role of MLN4924 in lipid metabolism. However, the means by which MLN4924 regulates the levels of fluphenazine needs further experimental validation. Another interesting metabolite found to be significantly upregulated in our data was b-GPA. A recent study has shown an association between increased basal metabolic rate and elevated production of b-GPA, implying its potential role in controlling obesity [33]. In addition, we identified high levels of uridine, a pyrimidine nucleoside that regulates metabolic processes through the activation of purinergic receptors [60,61]. Recent studies have emphasized the role of purinergic receptors in the regulation of adipocyte function and the pathogenesis of obesity [60,62,63]. Moreover, it is found that the liver is the predominant contributor to plasma uridine in the fed state, whereas the adipocyte dominates uridine biosynthetic activity in the fasted state [61]. The activation of the pyrimidine biosynthesis pathway has been verified to protect mice from obesity [64]. Overall, we not only identified novel metabolites regulated by MLN4924 treatment, but also found them to be lipid-metabolism-associated.

We showed that neddylation blockade by MLN4924 prevents HFD-induced hepatic steatosis of mice. This raised the possibility that NAFLD could be pharmacologically controlled through inhibition of the neddylation. Further research is required to uncover the role of MLN4924 in complexed metabolic processes in NAFLD and to study how to intervene in lipid metabolism disorders by speeding up the FAO process. A better understanding of the molecular mechanisms underlying NAFLD will provide an innovative platform for discovering more effective and tolerable anti-steatosis drugs.

## 5. Conclusions

In summary, we demonstrated that the neddylation blockade reduces the accumulation of lipid droplets, in part, through increasing mitochondrial FAO, which can stabilize liver function and attenuated liver structure pathological remodeling, thus postponing the progression of obesity and HFD-induced NAFLD. In addition, our results suggested that neddylation inhibition induced activation of FAO, which contributes to the pathological lightening of hepatic steatosis caused by lipid overload, and restrained the development of NAFLD. MLN4924 has potential therapeutic significance for treating NAFLD through the mitochondrial FAO pathway, and the neddylation process, in general, might be one of the most compelling targets for NAFLD treatment.

## Figures and Tables

**Figure 1 pharmaceutics-14-02460-f001:**
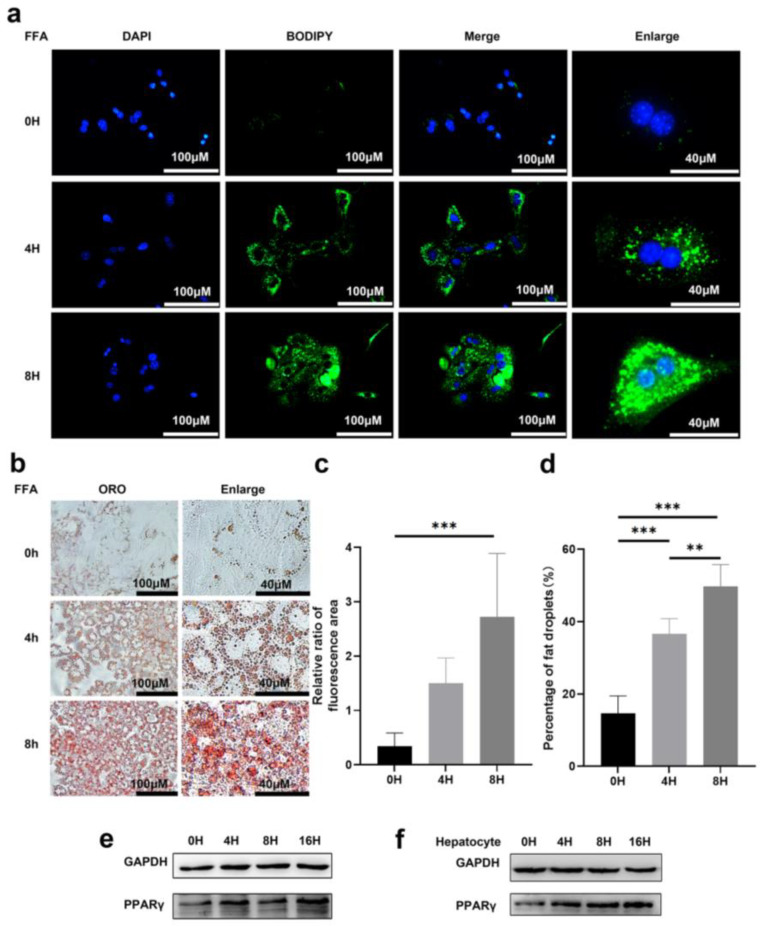
Lipid droplets can be induced by free fatty acid in hepatocytes. (**a**) Representative images of DAPI and BODIPY staining of primary hepatocytes treated with 1 mM FFA for indicated time. Scale bar: 40 μM and 100 μM. (**b**) Representative images of oil red O staining (ORO) of primary hepatocytes with indicated treatments. Scale bar: 40 μM and 100 μM. (**c**) Relative ratio of fluorescence area of DAPI and BODIPY staining were quantified. (**d**) Statistical analysis of area of fat droplets stained with oil red O (ORO) were presented. Huh7 cells (**e**) and primary hepatocytes (**f**) were treated with 1 mM FFA for 0, 4, 8, 16 h. Representative images of western blot showed the effects of FFA on PPARγ expression. GAPDH was used as an invariant control. Data represents the mean ± SD, *n* ≥ 3. One-way ANOVA was used for statistical analysis (** *p* < 0.01, *** *p* < 0.001).

**Figure 2 pharmaceutics-14-02460-f002:**
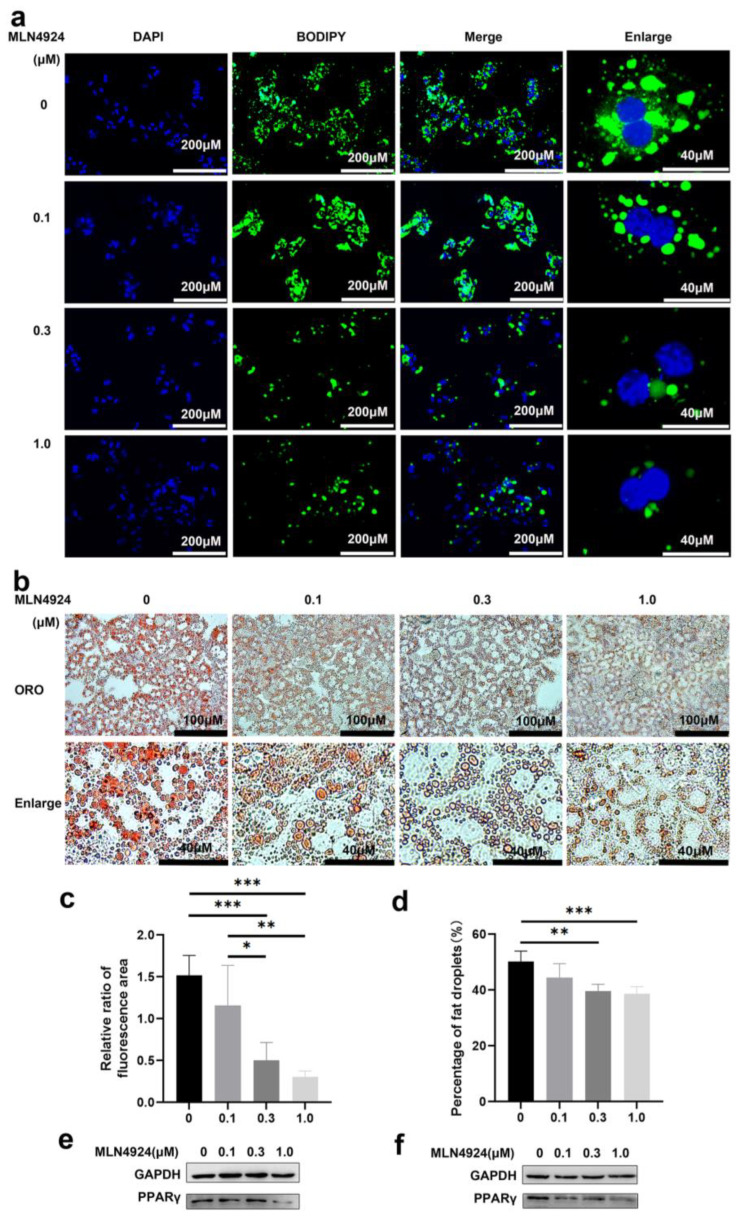
Neddylation inhibitor MLN4924 can inhibit lipid droplets induction by free fatty acid in hepatocytes. Primary hepatocytes were treated with 0, 0.1, 0.3 or 1 μM MLN4924 and 1 mM FFA for 8 h. Representative images of DAPI, BODIPY staining (**a**) and oil red O staining (ORO) (**b**) were presented. Scale bar: 40 μM and 200 μM for DAPI, BODIPY image, 40 μM and 100 μM for oil red O staining image. (**c**) Relative ratio of fluorescence area of DAPI and BODIPY staining were quantified. (**d**) Statistical analysis of area of fat droplets stained with oil red O (ORO) were presented. Huh7 cells (**e**) and primary hepatocytes (**f**) were treated with 0, 0.1, 0.3 or 1 μM MLN4924 and 1 mM FFA for 16 h. Representative images of western blot showed the effects of FFA on PPARγ expression. GAPDH was used as an invariant control. Data represents the mean ± SD, *n* ≥ 3. One-way ANOVA was used for statistical analysis (* *p* < 0.05, ** *p* < 0.01, *** *p* < 0.001).

**Figure 3 pharmaceutics-14-02460-f003:**
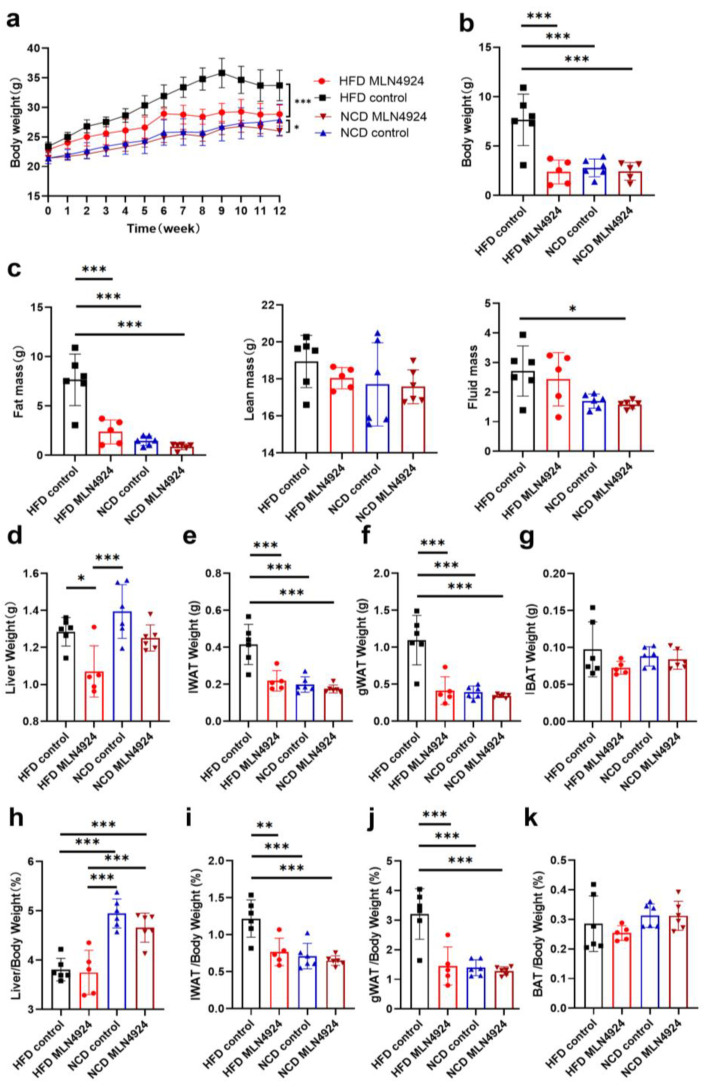
MLN4924 ameliorates fatty liver and obesity in mice. (**a**) Mice were divided into four groups: NCD Control mice, NCD MLN4924 mice, HFD Control mice, and HFD MLN4924 mice. All mice were weighed every week and the body weights (mean ± SD) are plotted as a function of time (*n* = 5–6 per group). (**b**) The body weight differences analysis of mice from four groups at the 12-week feeding endpoint (*n* = 5–6 per group). (**c**) Body composition analysis using MRI for four groups (*n* = 5–6 per group). (**d**–**g**) The weight of liver, iWAT, gWAT and BAT in mice of four groups were plotted. (**h**–**k**) The liver, iWAT, and gWAT and BAT weight percentage of body weight in mice from four groups were plotted (*n* = 5–6 per group). Data represents the mean ± SD, *n* ≥ 3. Two-way ANOVA was used for body weight analysis and one-way ANOVA was used for comparison among groups (* *p* < 0.05, ** *p* < 0.01, *** *p* < 0.001).

**Figure 4 pharmaceutics-14-02460-f004:**
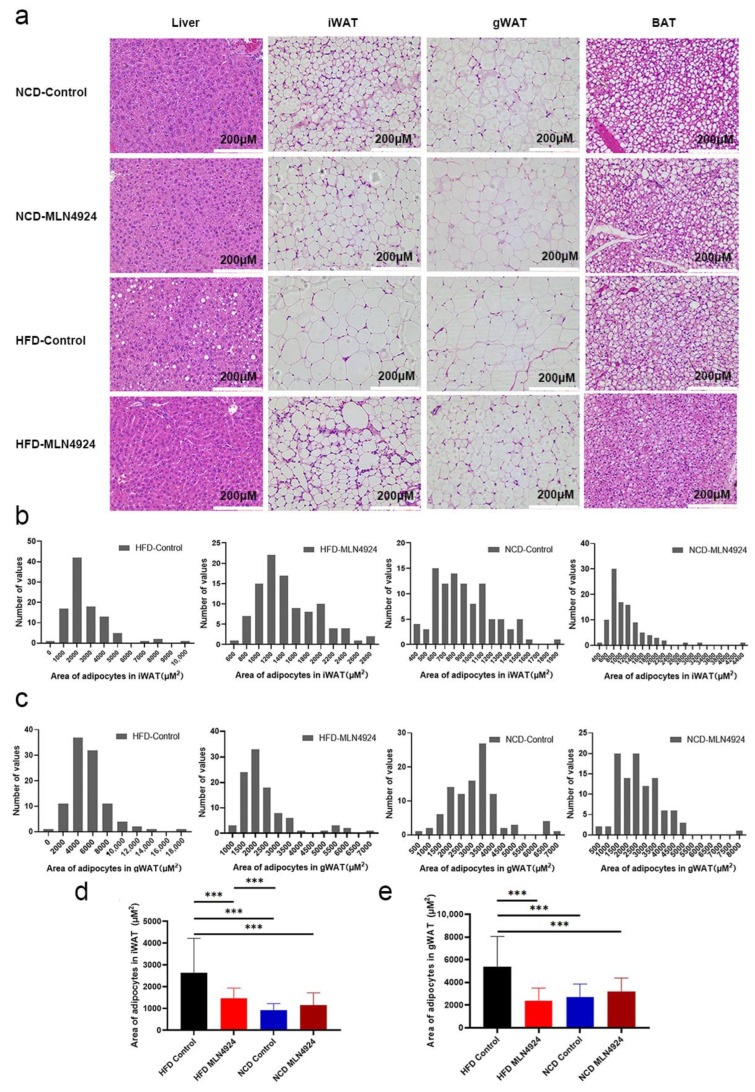
The generation and/or accumulation of lipid droplets in tissues of mice can be influenced by MLN4924. (**a**) Representative image of liver and adipose tissues and H & E staining are shown. (**b**,**c**) The statistical distribution map of adipocytes area of iWAT and gWAT in four groups were presented. (**d**,**e**) Statistical analysis of area of lipid droplets in iWAT and gWAT were presented. Data represents the mean ± SD, *n* ≥ 3. One-way ANOVA was used for statistical analysis (*** *p* < 0.001).

**Figure 5 pharmaceutics-14-02460-f005:**
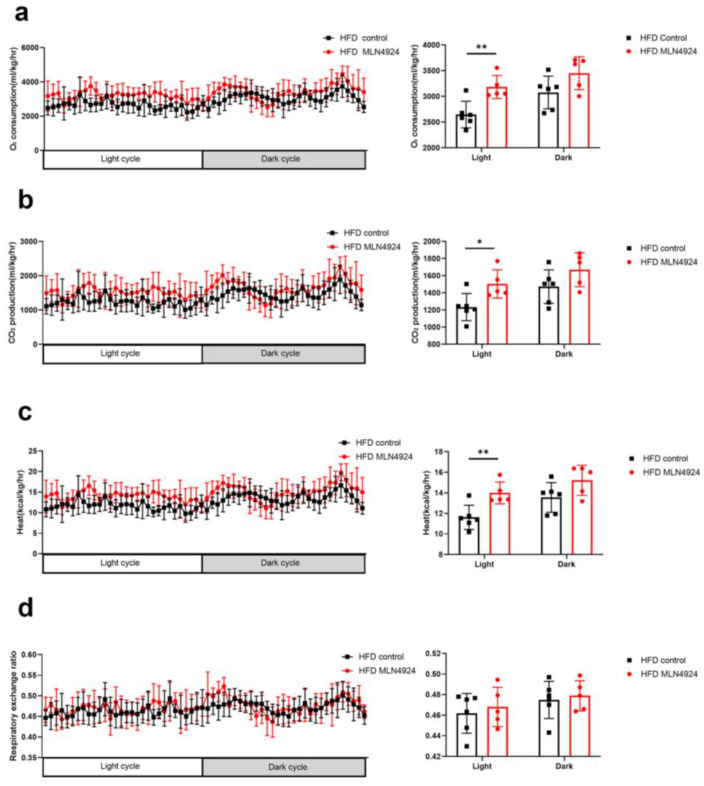
The metabolic conditions of mice can be influenced by MLN4924. (**a**–**d**) Mice fed with HFD were injected with or without MLN4924. Oxygen consumption, carbon dioxide generation, heat production and respiratory exchange ratio of two groups were measured (*n* = 5–6 per group). Data represents the mean ± SD, *n* ≥ 3. A two-tailed Student’s *t* test was used for statistical analysis (* *p* < 0.05, ** *p* < 0.01).

**Figure 6 pharmaceutics-14-02460-f006:**
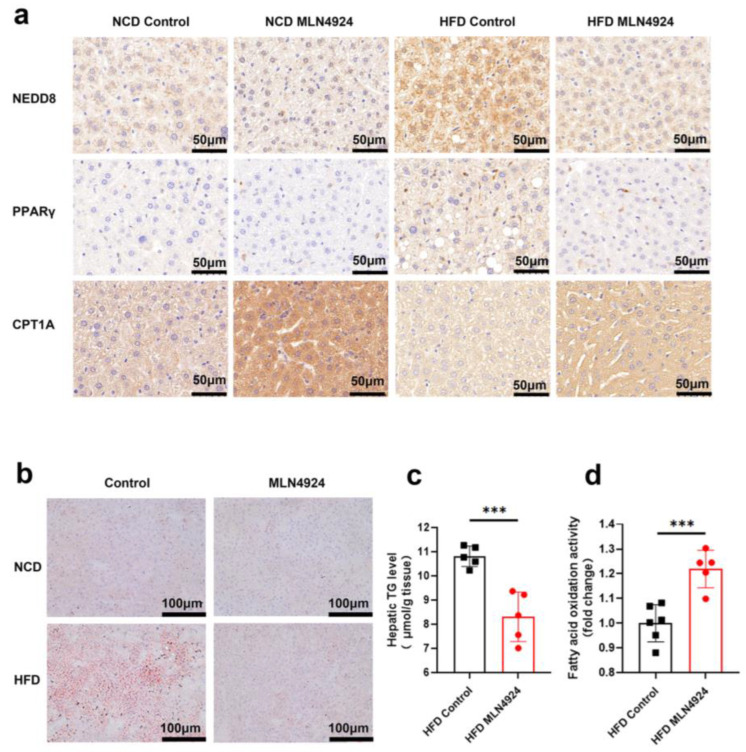
MLN4924 can regulate lipid metabolism in liver by increasing fatty acid oxidation and decreasing the triglycerides accumulation. (**a**) Representative immunohistochemical staining pictures of liver sections from NCD-fed and HFD-fed male mice treated with or without MLN4924 showed the expressions of NEDD8, PPARγ and CPT1A. (**b**) Representative images of ORO staining liver tissues in indicated mice were presented. (**c**) Hepatic triglyceride contents in HFD-fed mice were measured (*n* = 5 per group). (**d**) Mitochondria fatty acid oxidation fluxes in HFD-fed mice were measured (*n* =5–6 per group). Data represents the mean ± SD, *n* ≥ 3. A two-tailed Student’s *t* test was used for statistical analysis (*** *p* < 0.001).

**Figure 7 pharmaceutics-14-02460-f007:**
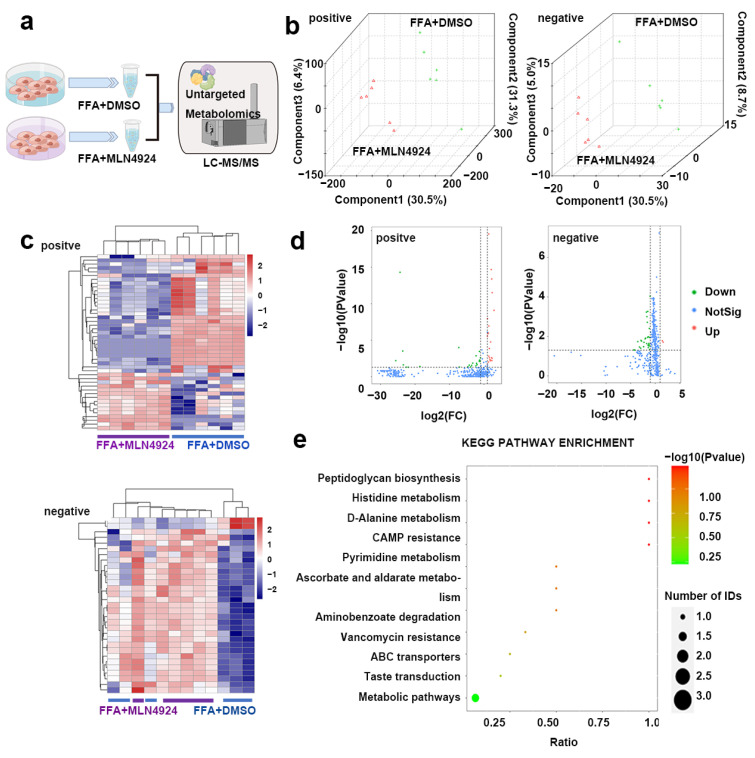
Lipidomics analysis on liver lipid metabolism after treatment of neddylation inhibitor MLN4924. (**a**) Huh7 cells were divided into FFA + DMSO and FFA + MLN4924 groups. LC-MS–based nontargeted metabolomic analysis detecting differential metabolites between two groups (*n* = 6 per group). (**b**) PLS-DA three-dimensional score plots for FFA + DMSO and FFA + MLN4924 groups. Images showed global sample distribution profiles analyzed by PLS-DA with the positive ion mode on the left and negative ion mode on the right. (**c**) Heatmap presented differentially expressed metabolites in FFA + DMSO group and FFA + MLN4924 group with the positive ion mode above and negative ion mode below. (**d**) Volcano plot showed lipidomics analysis of FFA + DMSO group and FFA + MLN4924 group with the positive ion mode on the left and negative ion mode on the right. The red dot represented the fold change (FC) > 2.0 and *p* value < 0.05. The green dot represents FC < 0.5, and *p* value < 0.05. (**e**) KEGG pathway enrichment analysis presented the main metabolic pathways identified between FFA + DMSO group and FFA + MLN4924 group by metabolites annotated to KEGG database.

**Table 1 pharmaceutics-14-02460-t001:** List of top differential metabolites found in FFA + DMSO treated group versus FFA + MLN4924 treated group (*p* < 0.05).

HMDB ID	log2fold	*p*-Value	Up/Down	*m*/*z*	Description
HMDB0001904	−22.012587	0.048762	Down	244.0920	3-Nitrotyrosine
HMDB0012243	−7.299020	0.000100	Down	395.1660	Kinetin-7-N-glucoside
HMDB0014592	−4.649321	0.042829	Down	352.2106	Dipivefrin
HMDB0013222	−3.586083	0.027941	Down	132.0766	Beta-Guanidinopropionic acid
HMDB0001212	−3.212516	0.034643	Down	217.0474	Hydantoin-5-propionic acid
HMDB0038968	−2.893064	0.015947	Down	281.0355	1-Propenyl 1-(1-propenylsulfinyl)propyl disulfide
HMDB0036650	−2.275692	0.019872	Down	325.0318	Sinalbin A
HMDB0015269	−2.131924	0.014434	Down	427.1073	Sulfinpyrazone
HMDB0034364	−1.790339	0.000586	Down	304.0628	Cepharadione A
HMDB0003040	−1.719765	0.023093	Down	267.0731	Arabinosylhypoxanthine
HMDB0041468	−1.685299	0.000997	Down	302.0658	1,3,5-Trihydroxy-10-methylacridone
HMDB0034364	−1.607278	0.001931	Down	304.0629	Cepharadione A
HMDB0000296	−1.520245	0.011211	Down	243.0616	Uridine
HMDB0014340	−1.508761	0.000539	Down	312.0947	Vidarabine
HMDB0032852	−1.483694	0.010759	Down	241.0825	2-(Ethylamino)-4,5-dihydroxybenzamide
HMDB0038968	−1.446022	0.027435	Down	281.0356	1-Propenyl 1-(1-propenylsulfinyl)propyl disulfide
HMDB0040795	1.033134	0.000000	Up	347.1854	7′-O-Methylmarmin
HMDB0057488	1.279045	0.010018	Up	1373.9427	CL(16:1(9Z)/16:1(9Z)/18:1(11Z)/16:1(9Z))
HMDB0040795	1.291746	0.000001	Up	347.1855	7′-O-Methylmarmin
HMDB0004886	1.389978	0.001551	Up	1158.7890	Trihexosylceramide (d18:1/24:0)
HMDB0014902	1.420620	0.000000	Up	444.1687	Mometasone
HMDB0009422	1.451045	0.016616	Up	810.5281	PE(20:4(8Z,11Z,14Z,17Z)/18:1(9Z))
HMDB0015621	1.562168	0.000135	Up	328.1063	Sulfadimethoxine
HMDB0012389	1.584051	0.002586	Up	812.5430	PS(18:1(9Z)/18:0)
HMDB0009090	1.610539	0.019622	Up	788.5444	PE(18:2(9Z,12Z)/18:0)
HMDB0010660	1.839512	0.004755	Up	762.5259	PG(18:3(6Z,9Z,12Z)/16:0)
HMDB0010661	1.893689	0.006705	Up	760.5106	PG(18:3(6Z,9Z,12Z)/16:1(9Z))
HMDB0010662	1.894613	0.005903	Up	790.5562	PG(18:3(6Z,9Z,12Z)/18:0)
HMDB0012378	2.180602	0.002500	Up	814.5566	PS(18:0/18:0)
HMDB0014761	2.250722	0.000000	Up	460.1635	Fluphenazine
HMDB0012356	2.319175	0.006175	Up	786.5241	PS(16:0/18:0)
HMDB0010624	2.361055	0.002972	Up	816.5743	PG(18:1(11Z)/20:3(8Z,11Z,14Z))

## Data Availability

The data presented in this study are available on reasonable request from the corresponding author.

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
