# Peer review of "MLN4924 Treatment Diminishes Excessive Lipid Storage in High-Fat Diet-Induced Non-Alcoholic Fatty Liver Disease (NAFLD) by Stimulating Hepatic Mitochondrial Fatty Acid Oxidation and Lipid Metabolites"

_pharmaceutics, 2022, doi:10.3390/pharmaceutics14112460_

Round 1

Reviewer 1 Report

The authors found that MLN4924 treatment diminishes excessive lipid storage in highfat diet-induced non-alcoholic fatty liver disease (NAFLD) by

stimulating hepatic mitochondrial fatty acid oxidation and lipid metabolites. Below are some comments:

1.    How did the authors demonstrate that MLN4924 treatment slow down the occurrence and development of NAFLD in mice from the study?

2.    Did the authors demonstrate that MLN4924 treatment compared to mock treatment improved the histological signature of NASH ( steatosis, balloon degeneration, liver inflammation, fibrosis, etc ) in HDF mices?

3.    Did MLN4924 treatment alter blood sugar homeostasis, lipid profile and body weight in mice?

4.    Will MLN4924 treatment shorten the life of mice?

5.    In a recent work in 2021, Serrano-Maciá, et al found that Neddylation inhibition ameliorates steatosis in NAFLD by boosting hepatic fatty acid oxidation via the DEPTOR-mTOR axis. What are the authors ‘ comments? What are the effects of MLN4924 on DEPTOR-mTOR axis in this study?

6.    Neddylation of sterol regulatory element-binding protein 1c is a potential therapeutic target for nonalcoholic fatty liver treatment. What are the effects MLN4924 on sterol regulatory element-binding protein 1c.

7.    Are all the signal proteins including PPARγ, . DEPTOR-mTOR axis or regulatory element-binding protein 1c significant in the heatmap shown in Fig 7?

Author Response

Dear Reviewers,
  Thank you for your comments concerning our manuscript entitled “MLN4924 treatment diminishes excessive lipid storage in high-fat diet-induced non-alcoholic fatty liver disease (NAFLD) by stimulating hepatic mitochondrial fatty acid oxidation and lipid metabolites” to PHARMACEUTICS (Manuscript ID:2001220 ). We highly appreciate the prompt review process and valuable comments from all the reviewers. We have revised our manuscript to our best efforts according to the reviewers’ suggestions. Our point-by-point responses to the questions raised by the reviewers and the revision of the manuscript are enclosed after the letter for your consideration.Your kind consideration will be greatly appreciated. I look forward to hearing your thoughts about this manuscript.

With best regards, 

Dongqin Yang Ph.D. 
Prof. Deparment of Digestive Diseases of Huashan Hospital of Fudan University 
Shanghai 200040, P. R. China 
E-mail: kobesakura@fudan.edu.cn 
Tel: 86-21-52888234 

Reviewer 2 Report

The manuscript “MLN4924 treatment diminishes excessive lipid storage in high- 2

fat diet-induced non-alcoholic fatty liver disease (NAFLD) by stimulating hepatic mitochondrial fatty acid oxidation and lipid metabolites” by Mengxiao Ge and collaborators was critically reviewed. 

The manuscript presents interesting data, but some concerns should be clarified. 

Although the study addresses the main research question in primary hepatocytes, with fascinating data, in the metabolomic studies, the authors used the Huh7 cancer cell line, which has been reported to present aberrant lipogenesis, a remarkable characteristic of HCC. The authors should carefully discuss this significant difference. 

Somehow, the authors must show changes in protein dennylation (Immunoblot), particularly those related to lipid homeostasis (perhaps PPARgamma, LXR, etc.), in order to be clear that it is undoubtedly due to this mechanism, which, although it can be inferred, must be demonstrated. 

Although the authors showed in figure 6A an IHC of Nedd8, this result should be precisely verified.

Author Response

(The authors gave the same response as above.)
